# Fast and Accurate Pose Estimation with Unknown Focal Length Using Line Correspondences

**DOI:** 10.3390/s22218253

**Published:** 2022-10-28

**Authors:** Kai Guo, Zhixiang Zhang, Zhongsen Zhang, Ye Tian, Honglin Chen

**Affiliations:** Northwest Institute of Nuclear Technology, Xi’an 710024, China

**Keywords:** pose estimation, line correspondences, unknown focal length, camera position, normal vector

## Abstract

Estimating camera pose is one of the key steps in computer vison, photogrammetry and SLAM (Simultaneous Localization and Mapping). It is mainly calculated based on the 2D–3D correspondences of features, including 2D–3D point and line correspondences. If a zoom lens is equipped, the focal length needs to be estimated simultaneously. In this paper, a new method of fast and accurate pose estimation with unknown focal length using two 2D–3D line correspondences and the camera position is proposed. Our core contribution is to convert the PnL (perspective-n-line) problem with 2D–3D line correspondences into an estimation problem with 3D–3D point correspondences. One 3D line and the camera position in the world frame can define a plane, the 2D line projection of the 3D line and the camera position in the camera frame can define another plane, and actually the two planes are the same plane, which is the key geometric characteristic in this paper’s estimation of focal length and pose. We establish the transform between the normal vectors of the two planes with this characteristic, and this transform can be regarded as the camera projection of a 3D point. Then, the pose estimation using 2D–3D line correspondences is converted into pose estimation using 3D–3D point correspondences in intermediate frames, and, lastly, pose estimation can be finished quickly. In addition, using the property whereby the angle between two planes is invariant in both the camera frame and world frame, we can estimate the camera focal length quickly and accurately. Experimental results show that our proposed method has good performance in numerical stability, noise sensitivity and computational speed with synthetic data and real scenarios, and has strong robustness to camera position noise.

## 1. Introduction

Camera pose estimation is an important step in computer vision, SLAM and photogrammetry [1,2,3,4,5], and the corresponding methods are mainly divided into two categories. The first category is based on deep learning using large training sets, which has become popular recently [6,7,8,9]. The second category is the traditional pose estimation method, which uses a small number of precise inputs for accurate estimation [10,11,12,13,14,15]. The former category is difficult to apply in high-precision measurements, and, in some real scenarios, such as weapon tests, it is difficult to obtain a large training set. The method proposed in this paper belongs to the latter, which uses a small number of precise inputs for pose estimation, and the most commonly used inputs here are 2D–3D point correspondences [16] and 2D–3D line correspondences [17]. Point and line features are common in real scenarios and are easy to obtain by humans; hence, there are many point-based pose estimation methods, called PnP (perspective-n-point) solvers [18,19,20], and line-based pose estimation methods, called PnL (perspective-n-line) solvers [21,22,23,24]. Moreover, there are some methods that estimate pose using both points and lines [25], but they are not applied widely. In addition, there are some specific methods using line correspondences that are parallel lines [26]. These parallel lines in the image plane would intersect at a point that is called the vanishing point. The specific methods seemingly use line correspondences, but, actually, we could also say that they use point correspondences (i.e., vanishing point). However, parallel lines do not always exist in real scenarios and they are also not applied widely, as with PnP or PnL solvers. Hence, here, we mainly introduce PnP and PnL solvers.

For PnP solvers, a 2D–3D point correspondence provides two constraints [27]. When all the intrinsic parameters are known and all the extrinsic parameters are unknown, at least three 2D–3D point correspondences are needed to estimate the camera pose, which is called a P3P solver [28], with up to four solutions, and an additional constraint (e.g., one more 2D–3D point correspondence) is needed to determine the unique solution. When there are four 2D–3D point correspondences, the camera pose and an intrinsic parameter (e.g., the focal length) can be simultaneously estimated, which is called P4Pf [29,30]. When five 2D–3D point correspondences are present, the camera pose and three intrinsic parameters (e.g., the focal length and radial distortion) can be simultaneously estimated, which is called P5Pfr [31]. All the above methods are nonlinear. When there are at least six 2D–3D point correspondences, the camera pose can be estimated linearly, which is called DLT (Direct Linear Transformation) [32,33]. In addition, there are some methods that use partial parameters of pose as prior knowledge, such as the known vertical direction [34,35,36] or camera position [37,38], and these methods can use fewer 2D–3D point correspondences, simplify the problem and increase the efficiency.

In some cases, no point features but line features are present. Similar to 2D–3D point correspondences, 2D–3D line correspondences are also widely used in pose estimation. The method proposed in this paper is an estimation method based on 2D–3D line correspondences. Similarly, a 2D–3D line correspondence can give two constraints. If all the intrinsic parameters are known and the six parameters of the pose are all unknown, at least three 2D–3D line correspondences are needed to estimate the pose, which is called P3L [39,40,41], and up to eight solutions can be obtained by solving nonlinear equations, and then another constraint is given to determine the unique solution. If some intrinsic parameters are unknown, more 2D–3D line correspondences are needed [42,43,44]. When there are four 2D–3D line correspondences, the focal length can be obtained, while the pose is estimated. When there are five 2D–3D line correspondences, the focal length and principal point (or radial distortion) can be obtained, while the pose is estimated. When there are at least six 2D–3D line correspondences, the camera pose can be linearly estimated [45]. As we can see, more parameters can be estimated using more 2D–3D line correspondences, and the computational complexity decreases (from nonlinear to linear estimation). However, sufficient 2D–3D line correspondences may not exist in the FOV (field of view), and a great deal of work is needed to measure the lines accurately. In this case, we need to finish the pose estimation with a smaller number of 2D–3D line correspondences, and this is the motivation for our work in this paper. If less than three 2D–3D line correspondences are required, some pose parameters need to be known. Some methods use IMU-based techniques to obtain the vertical direction [46]; that is, two angles are known, and then two 2D–3D line correspondences are used to estimate the pose. Similarly, based on this idea, we can measure the camera position in advance, which also reduces the number of 2D–3D line correspondences. With the development of technology, various tools are used to measure the camera position with high accuracy, such as total station and RTK (real-time kinematics). Therefore, in this paper, we propose a new method to simultaneously estimate the pose and focal length using two 2D–3D line correspondences and the camera position.

In this paper, two known 2D–3D line correspondences and the known camera position are used. In many cases, the camera position could be known as prior knowledge. In missile testing range and rocket launch applications, for example, the attitude measurement based on fixed cameras with a zoom lens is an important test. These cameras are fixed and the positions can be measured as prior knowledge by RTK or total station. The accuracy of the RTK and total station is high. In addition, with the growing prominence of the social security problem, VMCs (visual monitoring cameras) are used widely. In general, the position of the VMC is fixed and the lens orientation can be changed. In these cases, where the camera position is known, the presented method can be applied to estimate the focal length and pose using line correspondences. Each 3D line and camera position in the world frame can determine a plane, and then we can obtain two known planes in the world frame because two 3D lines are given. The angle between these two planes in the world frame can be calculated. Similarly, we can obtain two known planes in the camera frame because two 2D lines are given. Then, the angle with an unknown focal length, between these two planes in the camera frame, can be calculated. According to the principle of camera imaging, the two angles are equal and, consequently, an equation can be given with one parameter (i.e., focal length). Hence, the focal length can be estimated by using this geometric characteristic. In addition, the plane that contains the 3D line and camera position in the world frame, and the plane that contains the corresponding 2D line and camera position in the camera frame, are the same plane in space. Then, the unit normal vectors of the two planes can correspond through the rotation matrix, and, lastly, the unit normal vector correspondences are transformed into point correspondences under two intermediate frames with the same origin. Now, the pose estimation using line correspondences is transformed into pose estimation using point correspondences, and then we can efficiently estimate the camera pose using this characteristic. It can be seen that our proposed method uses only two 2D–3D line correspondences and the camera position to estimate the focal length and pose simultaneously, which improves the efficiency and expands the applicability of the proposed method while reducing the number of lines. Experimental results show that, compared with several existing pose estimation methods, our proposed method can achieve better performance in numerical stability, noise sensitivity and computational speed, both in synthetic data and real scenarios. In addition, our proposed method also has strong robustness to camera position noise.

The rest of this paper is as follows. Section 2 describes the materials and methods, and proposes the specific theoretical derivation of our proposed method. In Section 3, the numerical stability, noise sensitivity and computational speed of the proposed method and some other SOTA (state-of-the-art) methods are compared. Section 4 is the discussion; Section 5 is the conclusions.

## 2. Proposed Method

Two 3D lines *L*_1_, *L*_2_ and camera position *O_c_* are known in the world frame *S_w_*_1_ (*O_w_*_*X_w_Y_w_Z_w_*). The projections of the two 3D lines are denoted as *l*_1_, *l*_2_ on the image plane and *l*_1_, *l*_2_ are known in the camera frame *S_c_*_1_ (*O_c__X_c_Y_c_Z_c_*), as shown in Figure 1.

Line *L_i_* (*i* = 1, 2) and camera position *O_c_* can determine a plane and then we can obtain two planes that are denoted as π_1_, π_2_ in this paper. Because the lines and camera position are known in the world frame, the planes π_1_, π_2_ are known in the world frame and then we can obtain two unit normal vectors of the two planes, denoted as ***n_w_*_1_** and ***n_w_*_2_**, and they are known in the world frame *S_w_*_1_ (*O_w_*_*X_w_Y_w_Z_w_*). Similarly, by extracting two lines *l*_1_, *l*_2_ on the image plane, we can obtain two unit normal vectors ***n_c_*_1_** and ***n_c_*_2_** in the camera frame *S_c_*_1_ (*O_c__X_c_Y_c_Z_c_*), but both contain an unknown variable, which is the focal length. The angle between the two normal vectors in the camera frame is equal to the angle in the world frame. Therefore, an equation can be established based on this relation among the angles. The equation contains only one unknown variable, namely the focal length, and then the focal length can be solved.

In addition, a vector in the world frame can be transformed into another vector in the camera frame with the camera rotation matrix. Hence, the unit normal vector ***n_wi_*** in world frame *S_w_*_1_ (*O_w_*_*X_w_Y_w_Z_w_*) can be transformed into ***n_ci_*** in camera frame *S_c_*_1_ (*O_c__X_c_Y_c_Z_c_*) using ***n_c_*_i_** = ***Rn_wi_***; here, ***R*** is the camera rotation matrix, and contains all the pose information. According to this rotation relationship, we translate the origin of the world frame *S_w_*_1_ (*O_w_*_*X_w_Y_w_Z_w_*) to the camera position, and the new world frame *S_w_*_2_ (*O_w_*_2__*X_w_*_2_*Y_w_*_2_*Z_w_*_2_) is obtained. In this way, the projection relationship of unit normal vectors can be changed into the relationship between the point ***n_wi_*** in the world frame *S_w_*_2_ (*O_w_*_2__*X_w_*_2_*Y_w_*_2_*Z_w_*_2_), and the point ***n_ci_*** in the camera frame *S_c_*_1_ (*O_c__X_c_Y_c_Z_c_*) through the rotation matrix *R*. *R* here is the rotation matrix between world frame *S_w_*_2_ (*O_w_*_2__*X_w_*_2_*Y_w_*_2_*Z_w_*_2_) and camera frame *S_c_*_1_ (*O_c__X_c_Y_c_Z_c_*), and also the rotation matrix between world frame *S_w_*_1_ (*O_w_*_*X_w_Y_w_Z_w_*) and camera frame *S_c_*_1_ (*O_c__X_c_Y_c_Z_c_*). After this, the problem with line correspondences has been converted into the problem with point correspondences, which means that the PnL problem in this paper has changed into the PnP problem. It is a useful transformation that makes the calculation process fast and efficient. The specific calculation process is as follows.

### 2.1. Focal Length Estimation

The expression of *L_i_* is (***V_i_***, *P_i_*) in this paper, and ***V_i_*** is the unit direction vector of the line, while *P_i_* is an arbitrary point on the line. Then, *L_i_* in the world frame can be written as
(1)$$L_{i}=P_{i}+kV_{i}$$

Here, *k* is an arbitrary scale factor, unknown. Then, in the world frame, the unit normal vector $n_{wi}$ of the plane π***_i_*** passing through the line *L_i_* and the camera position *O_c_* is
(2)$$n_{wi}=\frac{V_{i}\times \vec{P_{i}O_{c}}}{\left\|V_{i}\times \vec{P_{i}O_{c}}\right\|}$$

The angle α between the unit normal vectors of the plane π_1_ and π_2_ can be obtained using
(3)$$\alpha =\arccos\frac{n_{w1}^{T}\times n_{w2}}{\left\|n_{w1}\right\|\cdot \left\|n_{w2}\right\|}$$

Assume that the pixel coordinates of the two endpoints of the line *l_i_* in the image plane of the corresponding *L_i_* are given as $\left(\begin{array}{cc} u_{2i-1} & v_{2i-1} \end{array}\right),\;\left(\begin{array}{cc} u_{2i} & v_{2i} \end{array}\right)$, which are known; the focal length of the camera is *f*, unknown; then, the normal vector of π_1_ and π_2_ in the camera frame is
(4)$$\begin{array}{l} n_{c1}=\left[\begin{array}{ccc} v_{1}-v_{2} & u_{2}-u_{1} & \frac{u_{1}v_{2}-u_{2}v_{1}}{f} \end{array}\right]=\left[\begin{array}{ccc} a_{1} & b_{1} & \frac{c_{1}}{f} \end{array}\right]\\ n_{c2}=\left[\begin{array}{ccc} v_{3}-v_{4} & u_{4}-u_{3} & \frac{u_{3}v_{4}-u_{4}v_{3}}{f} \end{array}\right]=\left[\begin{array}{ccc} a_{2} & b_{2} & \frac{c_{2}}{f} \end{array}\right] \end{array}$$

Since the angles between the normal vectors in the world frame and camera frame, respectively, are the same, then
(5)$$\cos\alpha =\frac{n_{c1}\cdot n_{c2}^{T}}{\left\|n_{c1}\right\|\left\|n_{c2}\right\|}=\frac{a_{1}a_{2}+b_{1}b_{2}+\frac{c_{1}c_{2}}{f^{2}}}{\sqrt{a_{1}^{2}+b_{1}^{2}+\frac{c_{1}^{2}}{f^{2}}}\sqrt{a_{2}^{2}+b_{2}^{2}+\frac{c_{2}^{2}}{f^{2}}}}=\frac{m_{1}f^{2}+m_{2}}{\sqrt{m_{3}f^{2}+c_{1}^{2}}\sqrt{m_{4}f^{2}+c_{2}^{2}}}$$

Let $\cos\alpha =m_{5}$ and then we can obtain a new equation
(6)$$\left(m_{3}m_{4}m_{5}^{2}-m_{1}^{2}\right)f^{4}+\left(m_{3}m_{5}^{2}c_{2}^{2}+m_{4}m_{5}^{2}c_{1}^{2}-2m_{1}m_{2}\right)f^{2}+m_{5}^{2}c_{1}^{2}c_{2}^{2}-m_{2}^{2}=0$$

In Equation (6), $f^{2}$ is regarded as a parameter, and the above equation is a quadratic equation of one variable $f^{2}$ with two solutions. According to the restriction that f>0 and $f^{2}>0$, the unique solution of focal length can be obtained. Now, the focal length estimation is finished. Then, the unit normal vectors in the camera frame can be given using
(7)$$\begin{array}{l} n_{c1}=\frac{n_{c1}}{\left\|n_{c1}\right\|}\\ n_{c2}=\frac{n_{c2}}{\left\|n_{c2}\right\|} \end{array}$$

Next, we will estimate the camera pose.

### 2.2. Pose Estimation

As can be seen from Section 2.1, the unit normal vectors $n_{w1},n_{w2},n_{c1},n_{c2}$ are all known at present, and then we can obtain
(8)$$\begin{array}{l} n_{c1}=R\cdot n_{w1}\\ n_{c2}=R\cdot n_{w2} \end{array}$$

Here, ***R*** is the rotation matrix between the world frame and camera frame. According to the camera projection relationship, a point *P_w_* in the world frame can be transformed to a point *P_c_* in the camera frame using
(9)$$P_{c}=R\cdot P_{w}+t$$

Here, *t* is the translation vector from the world frame to the camera frame. If we translate the origin of the world frame to the origin of the camera frame, *t* is zero and then
(10)$$P_{c}=R\cdot P_{w}$$

This is consistent with Equation (8) between the unit normal vectors. Inspired by this relationship, this paper regards the rotation relationship between the unit normal vectors as the rotation relationship between the points after the origins of the camera frame and the world frame coincide, and the coordinates of the corresponding points are the values of the unit normal vectors in the same frame, as shown in Figure 2.

Here, a new world frame *S_w_*_2_ (*O_w_*_2__*X_w_*_2_*Y_w_*_2_*Z_w_*_2_) is established after translating the origin of the original world frame *S_w_*_1_ (*O_w_*_*X_w_Y_w_Z_w_*) to the origin of the camera frame *S_c_*_1_ (*O_c__X_c_Y_c_Z_c_*). The relationship between the two world frames is as follows.
(11)$$S_{w2}=S_{w1}-O_{c}$$

Now, the two points *P_c_*_1_ and *P_c_*_2_ in the camera frame *S_c_*_1_ are identical to the two points *P_w_*_1_ and *P_w_*_2_ in the world frame *S_w_*_2_. Then, we establish another camera frame *S_c_*_2_ (*O_c_*_2__*X_c_*_2_*Y_c_*_2_*Z_c_*_2_) and world frame *S*_*w*3_ (*O_w_*_3__*X_w_*_3_*Y_w_*_3_*Z_w_*_3_), as shown in Figure 3.

The origin *O_c_*_2_ of the new camera frame *S_c_*_2_ (*O_c_*_2__*X_c_*_2_*Y_c_*_2_*Z_c_*_2_) is located at the camera position *O_c_*, and the unit direction vectors of each axis are calculated using
(12)$$\begin{array}{l} \vec{O_{c2}X_{c2}}=\frac{\vec{O_{c}P_{c1}}}{\left\|\vec{O_{c}P_{c1}}\right\|}\\ \vec{O_{c2}Z_{c2}}=\frac{\vec{O_{c2}X_{c2}}\times \vec{O_{c}P_{c2}}}{\left\|\vec{O_{c2}X_{c2}}\times \vec{O_{c}P_{c2}}\right\|}\\ \vec{O_{c2}Y_{c2}}=\vec{O_{c2}Z_{c2}}\times \vec{O_{c2}X_{c2}} \end{array}$$

Then, the camera frame *S_c_*_1_ can be transformed into the camera frame *S_c_*_2_ using
(13)$$\begin{array}{l} S_{c2}=T_{c\_c2}\cdot S_{c1}\\ T_{c\_c2}={\left[\begin{array}{ccc} \vec{O_{c2}X_{c2}} & \vec{O_{c2}Y_{c2}} & \vec{O_{c2}Z_{c2}} \end{array}\right]}^{T} \end{array}$$

Similarly, we establish the new world frame *S_w_*_3_ (*O_w_*_3__*X_w_*_3_*Y_w_*_3_*Z_w_*_3_), and the origin of the world frame *O_w_*_3_ is also located at the camera position *O_c_*. The unit direction vectors of each axis are calculated using
(14)$$\begin{array}{l} \vec{O_{w3}X_{w3}}=\frac{\vec{O_{c}P_{w1}}}{\left\|\vec{O_{c}P_{w1}}\right\|}\\ \vec{O_{w3}Z_{w3}}=\frac{\vec{O_{w3}X_{w3}}\times \vec{O_{c}P_{w2}}}{\left\|\vec{O_{w3}X_{w3}}\times \vec{O_{c}P_{w2}}\right\|}\\ \vec{O_{w3}Y_{w3}}=\vec{O_{w3}Z_{w3}}\times \vec{O_{w3}X_{w3}} \end{array}$$

Then, the world frame *S_w_*_2_ can be transformed into the world frame *S_w_*_3_ using
(15)$$\begin{array}{l} S_{w3}=T_{w2\_w3}\cdot S_{w2}\\ T_{w2\_w3}={\left[\begin{array}{ccc} \vec{O_{w3}X_{w3}} & \vec{O_{w3}Y_{w3}} & \vec{O_{w3}Z_{w3}} \end{array}\right]}^{T} \end{array}$$

Since the two points *P_c_*_1_ and *P_c_*_2_ in the camera frame are the same as the two points *P_w_*_1_ and *P_w_*_2_ in the world frame, camera frame *S_c_*_2_ (*O_c_*_2__*X_c_*_2_*Y_c_*_2_*Z_c_*_2_) and world frame *S_w_*_3_ (*O_w_*_3__*X_w_*_3_*Y_w_*_3_*Z_w_*_3_) are the same frame. Now, the transformations between all the frames are determined, as shown in Figure 4.

From Figure 4, we can obtain the relationship between the camera frame *S_c_*_1_ and the world frame *S_w_*_1_ using
(16)$$\begin{array}{l} S_{c1}=T_{w\_c}\cdot S_{w1}+t_{w\_c}\\ T_{w\_c}=T_{c\_c2}^{-1}\cdot T_{w2\_w3}\\ t_{w\_c}=-T_{c\_c2}^{-1}\cdot T_{w2\_w3}\cdot O_{c} \end{array}$$

Now, the pose estimation is finished.

## 3. Experiments and Results

First, the proposed method and some other existing state-of-the-art solvers (i.e., P3P [16], RPnP [13], GPnPf [47], DLT [48] and P3L [24] here) with excellent performance are comprehensively tested with synthetic data, including numerical stability, noise sensitivity and computational speed, in terms of the focal length and pose estimation.

Second, the proposed method in this paper uses the prior knowledge of the camera position, but the prior knowledge cannot be absolutely correct, and an error in the camera position may bring a large error to the final pose and focal length estimation. Therefore, here, we also need to analyze the robustness of the pose and focal length estimation method to camera position noise.

Finally, the proposed method and some other existing state-of-the-art solvers are indirectly tested with real images, which shows that the proposed method can also achieve good performance in the real scenarios.

### 3.1. Synthetic Data

We synthesize a virtual perspective camera with 1280 × 800 resolution. Its principal point is the center of the image, the pixel size is 14 μm, and the focal length is 50 mm. The camera is located at (2, 2, 2) in meters in the world frame. All the methods are tested on synthetic data with a perfect pinhole camera to ensure that the comparison in computational performance is fair. For our proposed method and the P3L solver, 2D–3D line correspondences are needed, and for the P3P, RPnP, GPnPf and DLT solvers, 2D–3D point correspondences are needed. Hence, 3D points and lines are placed in the FOV of the virtual camera. In this paper, three thousand 3D points and three thousand 3D lines (their lengths are all 5 m) are randomly generated in a box of [(−20, 20) × (−20, 20) × (180, 220) in meters in the world frame. Through the virtual camera, three thousand 2D points and three thousand 2D lines are generated. Then, we obtain the synthetic data, consisting of three thousand 2D–3D point correspondences and three thousand 2D–3D line correspondences. 

In this section, two 2D–3D line correspondences and three 2D–3D line correspondences from the synthetic data are randomly chosen for our proposed method and the P3L solver, respectively, for each trial. Three, four, five and six 2D–3D point correspondences from the synthetic data are randomly chosen for the P3P, GPnPf, RPnP and DLT solvers for each trial, respectively. Here, P3P, GPnPf, RPnP, DLT and P3L are used to estimate the pose that includes the orientation and translation, and the GPnPf is also used to estimate the focal length when we analyze the performance of our proposed method.

#### 3.1.1. Robustness to Camera Position Noise

The camera position is given by the RTK or total station in this paper, where the accuracy of RTK is better than 3 cm, and the accuracy of total station is better than 0.5 cm [49]. Therefore, the camera position has noise, which may affect the pose estimation. In this section, we need to analyze the robustness to the camera position noise, and then Gaussian noise, whose deviation level varies from 0 to 3 cm, is added to the camera position. Here, 10,000 random trials for each noise level are conducted independently, and the mean errors of rotation, translation, reprojection and focal length for each noise level are reported in Figure 5.

From Figure 5, it can be seen that as the camera position noise increases, so do the rotation error, the translation error, the reprojection error and the relative focal length. When the noise is 3 cm, the errors reach the maximum values, which are 0.08 degrees, 2.72 cm, 0.28 pixels and 0.008%, respectively. The results show that even if there are errors in rotation, translation, reprojection and focal length caused by camera position noise, they are still small, and we can say that our proposed method has strong robustness to camera position noise in terms of pose estimation.

#### 3.1.2. Numerical Stability

In this section, we test our proposed method in terms of numerical stability for pose estimation. In total, 10,000 independent trials are conducted using synthetic data with no noise added. We also compare our proposed method with P3P, RPnP, GPnPf, DLT and P3L, and the results are reported in Figure 6.

From Figure 6, it can be seen that all six methods have good numerical stability for pose estimation. Specifically, in terms of the rotation error, DLT obtains the best result, and our proposed method has the second. In terms of the translation error, our proposed method achieves the best and DLT achieve the second-best result. In terms of the reprojection error, P3P obtains the best result, RPnP obtains the second, DLT obtains the third and our proposed method obtains the fourth. As a whole, DLT has the best performance and our proposed method has the second-best performance for rotation, translation and reprojection.

Our proposed method also estimates the focal length, but the P3P, RPnP, DLT and P3L methods do not. Here, to analyze the numerical stability of the focal length estimation, we tested our proposed method compared with the GPnPf solver, which is one of the state-of-the-art solvers used to estimate the focal length, as shown in Figure 6 (bottom right). It can be seen that our proposed method and the GPnPf method both have good numerical stability, and our proposed method performs better than GPnPf. This result shows that our proposed method has better performance regarding numerical stability in terms of focal length estimation.

#### 3.1.3. Noise Sensitivity

In this section, we test our proposed method in terms of noise sensitivity using synthetic data with zero-mean Gaussian noise added onto the 2D points and lines. The noise deviation level varies from 0 to 1 pixel and 10,000 independent trials are conducted for each noise level. We also compare our proposed method with P3P, RPnP, GPnPf, DLT and P3L, and the mean errors of rotation, translation, reprojection and focal length are reported in Figure 7.

From Figure 7, it can be seen that all six methods have good noise sensitivity for pose estimation. Moreover, as the noise increases, so do the rotation error, the translation error, the reprojection error and the focal length error. Specifically, in terms of the rotation error, our proposed method, DLT and P3L perform similarly, and they achieve the best performance. GPnPf shows the second-best result. In terms of the translation error, our proposed method is the best and DLT is the second-best. In terms of the reprojection error, P3P has the best result, our proposed method has the second-best, and DLT is fourth. As a whole, our proposed method has the best performance for rotation, translation and reprojection.

Our proposed method also estimates the focal length, but the P3P, RPnP, DLT and P3L methods do not. Here, to analyze the noise sensitivity of focal length estimation, we tested our proposed method compared with the GPnPf solver, which is one of the state-of-the-art solvers used to estimate the focal length, as shown in Figure 7 (bottom right). It can be seen that our proposed method and the GPnPf method both have good noise sensitivity, and our proposed method performs better than GPnPf. This result shows that our proposed method has better performance regarding noise sensitivity in terms of focal length estimation.

#### 3.1.4. Computational Speed

Section 3.1.1, Section 3.1.2 and Section 3.1.3 show that our proposed method has good performance in terms of accuracy and stability. However, to fully evaluate a method, it is necessary to test the computational speed, because some methods achieve better results only when they use more complex computational processes, at the expense of computational speed. Therefore, we need to demonstrate the complexity of our proposed method by evaluating the computational speed.

In this section, 10,000 independent trials are conducted on a 3.3 GHz two-core laptop for all the methods, respectively, to test the computational speed. Then, the mean computational times are calculated, as reported in Table 1. 

From Table 1, it can be seen that our proposed method has the fastest computational speed according to the computational time. Specifically, the computational speed of our proposed method is 3.7 times, 4.4 times 15.0 times, 1.7 times and 3.4 times that of the latter five methods, respectively.

### 3.2. Real Images

In this section, our proposed method is indirectly tested with real images to show that it can work well with real scenarios. Simultaneously, we compare the performance of our proposed method with that of the other methods (i.e., P3P, RPnP, GPnPf, DLT and P3L solvers). 

In real scenarios, the ground truth of the camera pose is unknown and hence we cannot directly compare the value of the pose estimation with the ground truth. However, the positions of the spatial point and line are easily measured with high accuracy using some common tools (e.g., RTK, total station and ruler). Hence, the position can be used to test our proposed method indirectly in real images. In fact, pose estimation is preparation for practical applications, such as SfM, SLAM and photogrammetry. One important application is to measure the spatial positions of objects in photogrammetry, where two or more cameras after pose estimation are used to finish the measurement, namely stereo vision. It can be seen that the accuracy of the position of the point is affected by the pose estimation and, hence, the accuracy of the position can reflect the accuracy of pose estimation. 

Here, two rectangular boxes are placed on a checkerboard, whose sizes are known. Many points and lines exist in this scenario, and their positions are all known as ground truth. Two mobile phone cameras are used to capture real images from two different perspectives, as shown in Figure 8. 

The focal length of the camera is 4.71 mm, the resolution is 4000 × 3000, and the pixel size is 1.6μm. The positions of the two cameras are given by the total station. Then, two lines are randomly chosen to estimate the pose for our proposed method, three lines for the P3L method, three points for the P3P method, four points for the GPnPf method, five points for the RPnP method and six points for the DLT method. After pose estimation using the six methods, the positions of the remaining points can be given as measured values by stereo vision [50]. Here, the measured values are denoted as $P_{i},\left(1\leq i\leq n\right)$, where *n* is the number of remaining points for the checkerboard. Because the size of the checkerboard is known, the positions of these points are known as the ground truths, which are denoted as $P_{i}^{\prime }$. We use the mean relative position errors $E_{position}$, between the measured value and the ground truth, to indicate the accuracy of all six methods, and the formula is written as
(17)$$E_{position}=\frac{\sum_{i=1}^{n}\frac{\left|P_{i}-P_{i}^{\prime }\right|}{P_{i}^{\prime }}}{n}$$

The mean relative position errors between the measured value and the ground truth are reported in Table 2.

From Table 2, it can be seen that our proposed method has the lowest error, the P3P method has the second-lowest and the RPnP method has the highest. This is consistent with the results for the synthetic data and shows that our proposed method can work well with synthetic data and real images.

In addition, we also test our proposed method in terms of computational speed in real images, and the experiment is conducted on a 3.3 GHz two-core laptop for all the methods. Specifically, the mean computational speed of our proposed method is 3.4 times, 3.9 times 16.2 times, 1.5 times and 3.7 times that of the latter five methods, respectively. This is basically consistent with the results for the synthetic data.

Last, we compare the projection of the known 3D line using the estimated pose to the corresponding line segment on the real images, and then we can intuitively observe whether they match, as shown in Figure 9. 

The projection is affected by the pose estimation, and, hence, the error between the projection and the corresponding line segment on the real images can intuitively reflect the error of pose estimation. From Figure 9, it can be seen that the error between the projection and the corresponding line segment on the real images is low, which shows, from another perspective, that our proposed method has good performance in real scenarios. 

## 4. Discussion

In this paper, we propose a fast and accurate method to estimate the focal length and pose using two 2D–3D line correspondences and the camera position. Our core contribution is to convert the PnL problem with 2D–3D line correspondences into the estimation problem with 3D–3D point correspondences. To the best of our knowledge, this is the first study to use the camera position and line correspondences simultaneously to estimate the focal length and pose. Using the camera position as the prior knowledge, two planes can be obtained in the camera frame and world frame, respectively. Then, two angles between the two planes in the camera frame and world frame, respectively, can be obtained. Since there is only a rigid body transformation, the angle remains the same in either frame. Hence, the two angles are equal, and, using this information, we can estimate the focal length efficiently and independently.

When we estimate the camera pose, another geometric characteristic is used, where we establish the transform between the unit normal vectors of the two planes with this characteristic, and this transform can be regarded as the camera projection of the 3D point. Then, the pose estimation using line correspondences is converted into pose estimation using 3D–3D point correspondences in intermediate frames, and the latter pose estimation can be finished quickly using traditional point-based solvers. The differences and advantages of our proposed method are discussed as follows.

### 4.1. Differences and Advantages

Compared with other existing methods using 2D–3D line correspondences, the first difference is to use the camera position as prior knowledge, which can simplify the problem of the focal length and pose estimation. The known camera position can reduce the number of 2D–3D line correspondences, improve the efficiency and expand the applicability. The real scene might not have sufficient line correspondences, and, in this case, our proposed method, with fewer 2D–3D line correspondences, can work better with the scenario. In addition, if the scene has mass line correspondences and outliers exist, generally, the RANSAC (RANdom SAmple Consensus) is used to eliminate the outliers. The RANSAC has a key parameter, which is the minimal set of correspondences for estimating pose. The smaller the minimal set, the faster the computational speed of eliminating the outliers. Hence, the RANSAC with our proposed method is faster than that with other existing methods when the scene has mass line correspondences and outliers exist. 

When the focal length is estimated, only a quadratic equation of one variable is used. We can solve the focal length directly, not using iterations and nonlinear algorithms, which is the main reason that our proposed method has better numerical stability and noise sensitivity in terms of focal length estimation. In addition, using the characteristic whereby the focal length is greater than zero, a single solution can be obtained, and unlike some existing methods, no other constraint is needed. The two characteristics described above are one reason that our proposed method has faster computational speed. When the camera pose is estimated, the main calculation process only involves multiplication, division and matrix operation, and there is also no nonlinear computation. This is the main reason that our proposed method has good performance in terms of numerical stability, noise sensitivity and computational speed in terms of pose estimation. 

In addition, because our proposed method does not involve nonlinear computation and iteration, there is no multi-solution phenomenon. This is another reason that our proposed method performs better than the other existing methods in terms of computational speed, as described in Section 3.1.4. It can be also seen that our proposed method and the DLT solver perform best in terms of numerical stability, noise sensitivity and computational speed compared to the other four methods. The main reason is that our proposed method and the DLT solver do not both involve nonlinear computation, but the other four methods do.

The level of simplification using prior knowledge is different, which means that the benefit might be high or low for different methods. As described in Section 3.1.1, our proposed method has good robustness to camera position noise, and this means that any error in the camera position does not bring a significant error for focal length and pose estimation. This is also a reason that our proposed method has higher accuracy.

Briefly, the proposed method has the following differences and advantages. (1) It is the first method to use the camera position and only two 2D–3D line correspondences simultaneously to estimate the focal length and pose; (2) it has strong robustness to prior knowledge, i.e., the known camera position; (3) no multi-solution phenomenon exists both for focal length and pose estimation; (4) it has better performance in terms of numerical stability, noise sensitivity and computational speed; (5) it can work well with synthetic data and real scenarios. However, the main disadvantage is that the camera position needs to be known in advance and many cases might not have the necessary conditions to measure the camera position.

### 4.2. Future Work

In the future, we will extend the idea in this paper to continue two main tasks. First, we will use more 2D–3D line correspondences and known camera positions to estimate more intrinsic parameters (e.g., radial distortion), rather than only the focal length and pose, which will expand the usable range of our proposed method. Second, as described in Section 4.1, the RANSAC + our proposed method will be used to estimate the camera pose from mass 2D–3D line correspondences, even if the line correspondences have some outliers. Our ultimate goal is to use RANSAC + our proposed method to work with real scenarios, e.g., SfM and SLAM. In addition, our method and the other methods in this paper used good lines. If imperfect lines exist, we could use the combination of a line detection algorithm and manual intervention to further extract and optimize the imperfect lines. This might be done in the future.

## 5. Conclusions

We proposed a fast and accurate method to estimate the focal length and pose using only two 2D–3D line correspondences and the camera position. The geometric characteristic whereby the angle between two planes is not changed in different frames is used to estimate the focal length. Then, another geometric characteristic, whereby the pose estimation using 2D–3D line correspondences in this paper can be converted into pose estimation using 3D–3D point correspondences in intermediate frames, is used to estimate the pose. The two calculation processes do not involve nonlinear computation and both have no multi-solution phenomenon. Experimental results show that our proposed method has better performance in terms of numerical stability, noise sensitivity and computational speed in synthetic data and real images compared to several state-of-the-art pose estimation solvers.

## Figures and Tables

**Figure 1 sensors-22-08253-f001:**
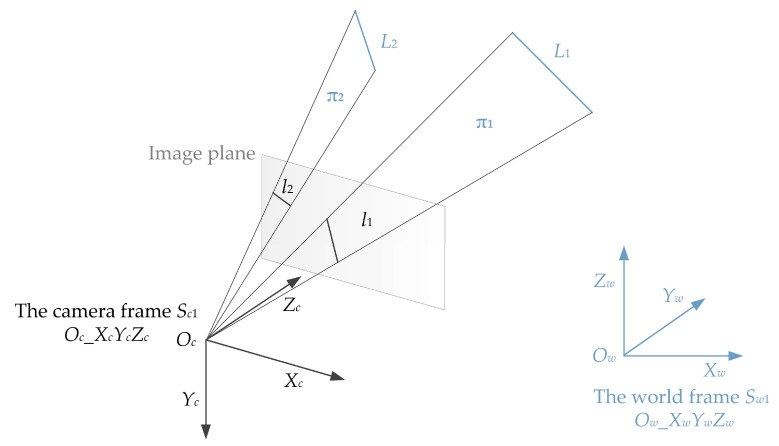
Two 3D lines and their projections in our proposed method.

**Figure 2 sensors-22-08253-f002:**
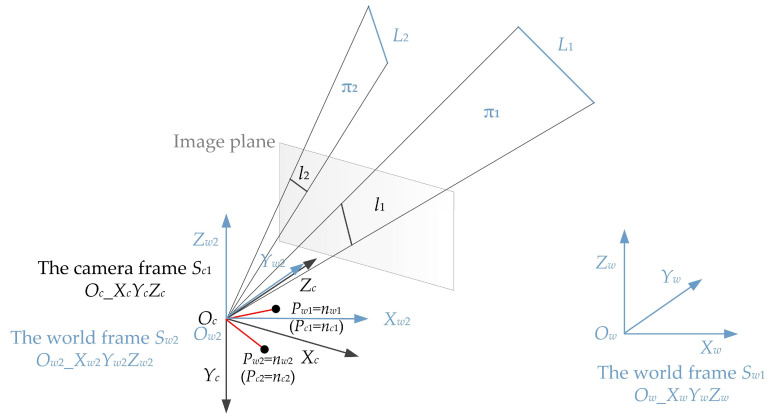
The geometry of our proposed method for pose estimation.

**Figure 3 sensors-22-08253-f003:**
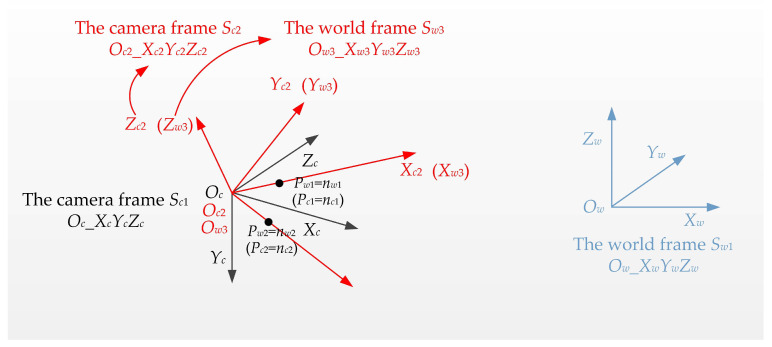
New camera frame and world frame plotted in red.

**Figure 4 sensors-22-08253-f004:**
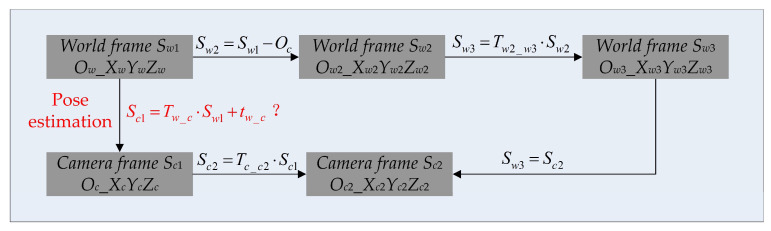
The transformations between each pair of frames. The transformation written in red is what needs to be solved in this paper, and this transformation contains all the pose information.

**Figure 5 sensors-22-08253-f005:**
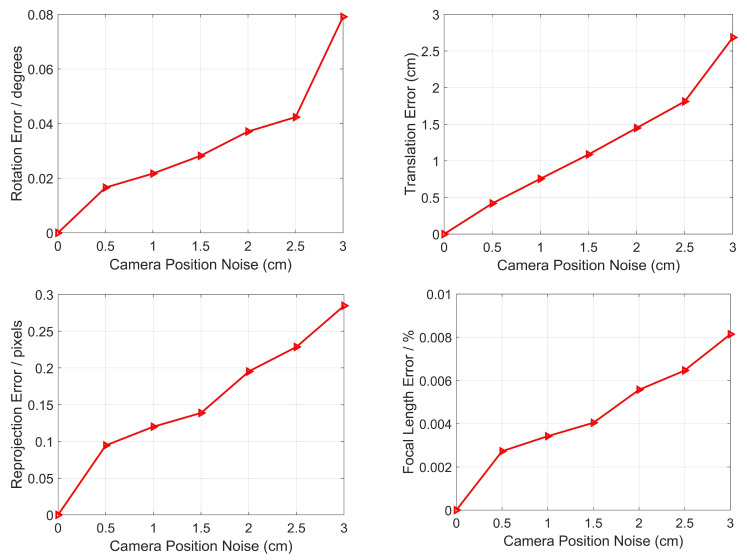
Robustness to camera position noise for the rotation error (**top left**), the translation error (**top right**), the reprojection error (**bottom left**) and the relative focal length (**bottom right**).

**Figure 6 sensors-22-08253-f006:**
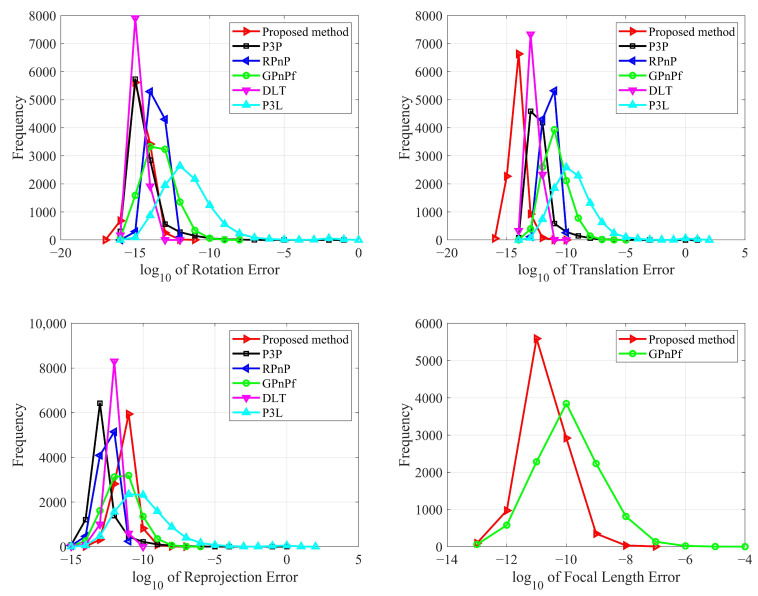
Numerical stability for our proposed method (Red), P3P (Black), RPnP (Blue), GPnPf (Green), DLT (Purple) and P3L (Cyan). The top left is the rotation error, the top right is the translation error, the bottom left is the reprojection error, and the bottom right is the focal length error.

**Figure 7 sensors-22-08253-f007:**
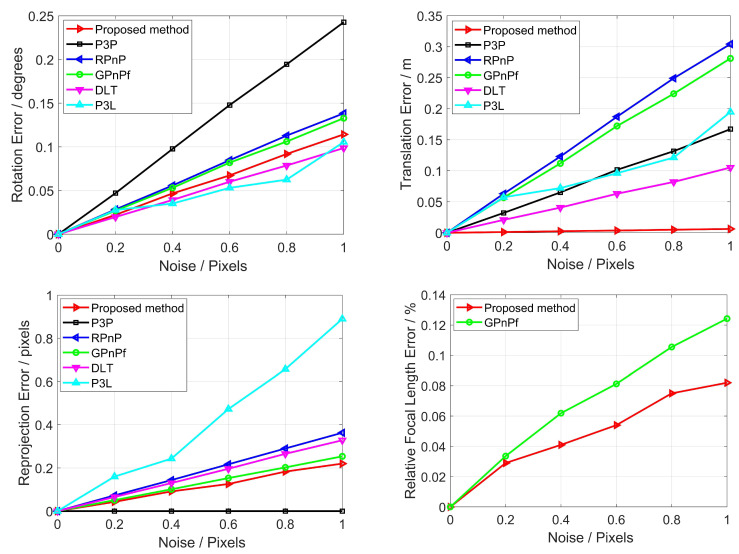
Noise sensitivity for our proposed method (Red), P3P (Black), RPnP (Blue), GPnPf (Green), DLT (Purple) and P3L (Cyan). The top left is the rotation error, the top right is the translation error, the bottom left is the reprojection error, and the bottom right is the focal length error.

**Figure 8 sensors-22-08253-f008:**
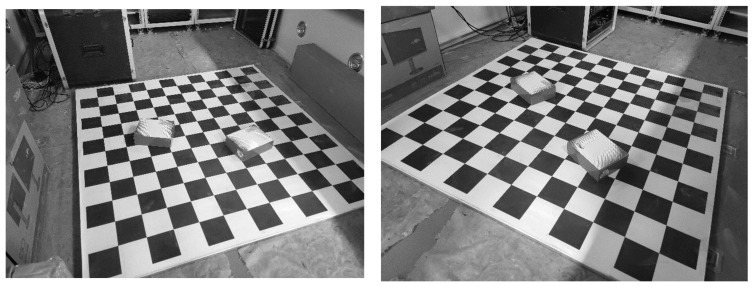
Real images captured from two different perspectives. Two rectangular boxes are placed on a checkerboard; the sizes of the boxes and checkerboard are known.

**Figure 9 sensors-22-08253-f009:**
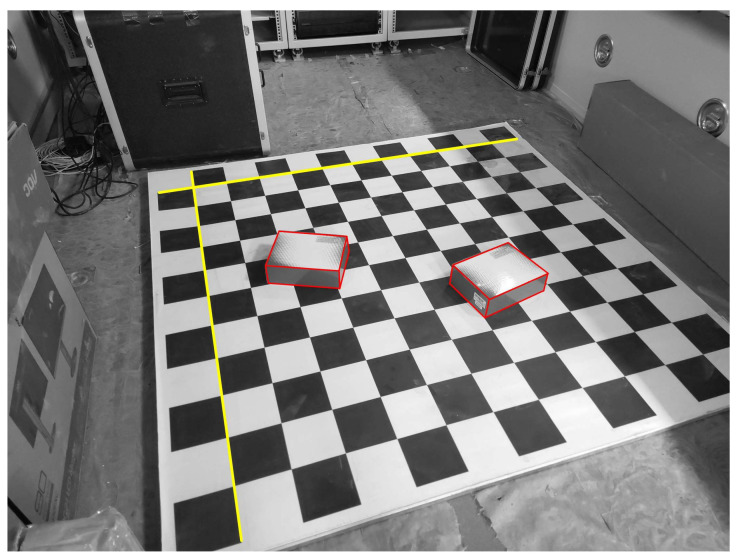
The projections (Red) of the known lines using the estimated pose with two 3D lines (Yellow).

**Table 1 sensors-22-08253-t001:** Computational time.

Method	Our Proposed Method	P3P	RPnP	GPnPf	DLT	P3L
Computational time	0.46 ms	1.72 ms	2.03 ms	6.88 ms	0.77 ms	1.56 ms

**Table 2 sensors-22-08253-t002:** Relative errors of position.

Method	Our Proposed Method	P3P	RPnP	GPnPf	DLT	P3L
Mean relative error %	0.43	0.51	1.79	1.06	0.56	0.62

## Data Availability

The data presented in this study are available in the manuscript.

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
