# Peer review of "Fast and Accurate Pose Estimation with Unknown Focal Length Using Line Correspondences"

_sensors, 2022, doi:10.3390/s22218253_

Round 1

Reviewer 1 Report

Dear authors,

thank you very much for the manuscript. All the descriptions and figures are detailed and clear and you had quite some work with the data analysis. My very few remarks are fed into the attached pdf-document.

The only point im struggling with, is the overall scientific value of the idea. I don't know of any case where the camera position (extrinsics) are known a priori. Especially not when accuracy is neccessary. That is probably the point why nobody has cared about such until now. But maybe my own expertise is just limited to few situations...

So I leave it to the other reviewers or the editor to put a final decision on the publication of your manuscript.

But - don't get me wrong - the manuscript itself and the research is done in a consistent and convenient way.

Best Regards

Reviewer 2 Report

Authors proposed an analytical solution for camera pose and focal length estimation under known camera positions, and two 2D-3D lines structures. The overall structure is clear and well established. And this technical field could be categorized as camera calibration. From the definition of manuscript, there are totally 4 degrees of freedom need to be determined under several known cues. From section 2.1 and 2.2, proposed method looks likely an analytical solution depending on two “normal vectors” of pi-planes. As a result, by neglecting numerical error, the error should be as small as zero. I’m wondering which factor was “optimized” during the overall pipeline?

In accuracy estimation:

The focal length is assumed as a simplified factor of the intrinsic parameter. It seems the accuracy strongly relies on the correctness of camera position and two 3D line structures. I’m not totally convinced why a simplified parameter (only one f instead of 4~5 factors in intrinsic matrix) would have better accuracy than previous state-of-the-art methods. In synthetic data, do authors implement the algorithm of line detection (ex. edge detection or subpixel estimation), and what about the situation of an imperfect line?

Computational performance (of computer synthetic cases):

In my opinion, the comparison in computational performance is not quite fair. The main reason is that the camera is assumed as a perfect pinhole camera. That means the difference of focal lengths along x and y directions is neglected, no distortion effect is involved, and a perfect central projection is given. Such an ideal case will no doubt play better performance.

Recommendation:

Notation: The statement and notation in section 2.1 and 2.2 are mess and difficult to understand. If possible, please list a table for briefly explanation. Several writing formats look informal and like a programming statement. For example, matrixes and vectors are sometime confusing in multiplication.

Titles in Figure: please use “(cm)” to replace “/cm”. The notation may cause confusion.

English: There are several typos (such as at line185, 357), and English of manuscript needs to proof.

Round 2

Reviewer 2 Report

The letter of authors' response answers most of my questions. I have no extra question regarding the method and structure.